# Multifunctional RNA-binding proteins influence mRNA abundance and translational efficiency of distinct sets of target genes

**Valentin Schneider-Lunitz**[1‡], **Jorge Ruiz-Orera**[1‡], **Norbert Hubner**[1,2,3]*, **Sebastiaan van Heesch**[4]*

**1** Cardiovascular and Metabolic Sciences, Max Delbrück Center for Molecular Medicine in the Helmholtz Association (MDC), Berlin, Germany, **2** Charité-Universitätsmedizin, Berlin, Germany, **3** DZHK (German Centre for Cardiovascular Research), Partner Site Berlin, Berlin, Germany, **4** Princess Máxima Center for Pediatric Oncology, Utrecht, The Netherlands

‡ Co-first authors
* nhuebner@mdc-berlin.de (NH); s.vanheesch@prinsesmaximacentrum.nl (SvH)

**Data Availability Statement:** This study includes no newly generated data deposited in external repositories. Accession numbers for all data used in this study have been reported within the

## Abstract

RNA-binding proteins (RBPs) can regulate more than a single aspect of RNA metabolism. We searched for such previously undiscovered multifunctionality within a set of 143 RBPs, by defining the predictive value of RBP abundance for the transcription and translation levels of known RBP target genes across 80 human hearts. This led us to newly associate 27 RBPs with cardiac translational regulation *in vivo*. Of these, 21 impacted both RNA expression and translation, albeit for virtually independent sets of target genes. We highlight a subset of these, including G3BP1, PUM1, UCHL5, and DDX3X, where dual regulation is achieved through differential affinity for target length, by which separate biological processes are controlled. Like the RNA helicase DDX3X, the known splicing factors EFTUD2 and PRPF8—all identified as multifunctional RBPs by our analysis—selectively influence target translation rates depending on 5' UTR structure. Our analyses identify dozens of RBPs as being multifunctional and pinpoint potential novel regulators of translation, postulating unanticipated complexity of protein-RNA interactions at consecutive stages of gene expression.

## Author summary

The lifecycle of an RNA molecule is controlled by hundreds of proteins that can bind RNA, also known as RNA-binding proteins (RBPs). These proteins recognize landing sites within the RNA and guide the RNA's transcription from DNA, its processing into a mature messenger RNA, its translation into protein, or its degradation once the RNA is no longer needed. Although we now mechanistically understand how certain RBPs regulate these processes, for many RBP-target interactions the consequences imposed by RNA binding are not well understood. For 143 RBPs with known RNA binding positions, the authors of the current study investigated how RNA molecules responded to fluctuations in the expression levels of these RBPs, across each of 80 human hearts. Using statistical

Materials and Methods section with reference to the studies that generated and published the data. All scripts generated for the analyses are available via the GitHub development platform at: https://github.com/vschnei/Dual-function_RBP_manuscript_analysis.

**Funding:** This work was supported by the European Union's Horizon 2020 research and innovation programme [European Research Council (ERC) advanced grant, grant agreement n° AdG788970 to N.H.; https://erc.europa.eu]; the Leducq Foundation [11 CVD-01 to N.H.; https://www.fondationleducq.org]; and the "Bundesministerium für Bildung und Forschung" [grant CaRNAtion to N.H; https://www.dfg.de]. The funders had no role in study design, data collection and analysis, decision to publish, or preparation of the manuscript.

**Competing interests:** The authors have declared that no competing interests exist.

approaches, they could show that many RBPs influenced stages of the RNA lifecycle that they were not known to be involved in. Some RBPs turned out to be true "all-rounders" of RNA metabolism: they controlled the RNA transcript levels of some genes, whereas they influenced the translation rates of others. This unexpected multifunctionality unveiled previously hidden aspects of the everyday RNA-binding protein working life.

## Introduction

RNA-protein interactions are crucial for a wide range of processes in multiple subcellular compartments, including RNA transcription, splicing, editing, transport, stability, localization, and translation [1]. Using state-of-the-art mass spectrometry-based approaches, recent studies have identified up to thousands of potential RNA-binding proteins (RBPs), although for many their precise roles remain unknown [2–4]. Whereas RBPs can interact with target RNAs through well-defined protein-RNA binding domains (RBD), uncharacterized RBDs are frequently discovered [5,6] highlighting the complex, diverse, and still largely unknown nature of RNA-protein interactions. The possession of more than a single RBD could theoretically allow RBPs to be multifunctional, for instance through the separate regulation of different sets of targets [7]. Multifunctionality may additionally be established through the condition- or cell type-specific expression of RBPs and their interaction partners, or through dynamic shuttling of RBPs between different subcellular compartments, such as the nucleus and cytosol [8–11]. Accordingly, out of a large number of RBPs whose subcellular localization was recently evaluated [12], the vast majority could be detected in more than one compartment, albeit of thus far largely unknown biological significance. The potential importance of RBP shuttling was exemplified by the subcellular redistribution of dozens of RBPs upon global induction of mRNA decay [13], and compartment-specific changes in RBP interactomes that could be witnessed upon cellular stress [14]. These observations indicated that RBPs may dynamically relocate to form a mechanistic bridge between normally compartment-restricted consecutive stages of gene expression regulation: some taking place in the nucleus (e.g., transcription, mRNA processing, splicing), others in the cytosol (e.g., translation, RNA decay).

RBPs are crucial regulators of mRNA translation—the synthesis of protein from a messenger RNA template by ribosomes [15]. In the cytosol, RBPs can act in a general capacity (e.g., mRNA translation initiation or elongation), as well as in more specialized functions where select accessory RBPs or heterogeneous ribosomes facilitate the translation of dedicated subsets of target RNAs [16–18]. In the nucleus, RBPs can coordinate the process of splicing and translation by modulating different structural and sequence properties of the mRNAs that are exported to the cytoplasm. For instance, highly structured regions in 5' UTRs can decrease the efficiency of translation initiation at the cost of an overall lower translational output [19–22], whereas increased RNA structures in CDS or 3' UTR regions may also enhance transcript stability and hence mRNA half-life, in turn yielding higher protein output over time [23,24]. Additionally, alternative UTR usage can expose translated upstream ORFs (uORFs) or miRNA binding sites affecting mRNA translation and/or stability. Other alternative splice events might affect intrinsic properties of an mRNA, such as the 'swiftness' of nuclear export [25], codon usage [26,27], affinity with the exon junction complex [28], or susceptibility to nonsense-mediated decay [29].

Mechanistic insight into the quantitative effects of RBP expression on mRNA translation– i.e., when translation rates of the endogenous RBP targets respond directly to changes in RBP abundance–was recently provided by several studies [30–32]. These studies used this

quantitative relationship to assign novel functions to known RBPs and to investigate the kinetics through which RBPs regulate their targets. For instance, Chothani *et al.* used a computational correlation approach to define the frequency with which RBP levels correlate with target translation rates. This helped to pinpoint key RBP network hubs that were crucial for translational regulation during cardiac fibrosis *in vitro* and *in vivo* [30]. In a separate study, Luo *et al.* used luciferase-based 3' UTR tethered function assays for 690 RBPs, identifying 50 RBPs whose expression induced significant positive or negative effects on mRNA stability and/or translation. This resulted in the novel characterization of the stress granule RBP UBAP2L as a ribosome-associated RBP [31]. Lastly, Sharma *et al.* developed a methodology to investigate RNA-protein kinetics in a time-resolved manner for the RBP DAZL [32]. This approach helped establish a quantitative relationship that precisely explained the effect of DAZL on mRNA levels and ribosome association, which correlated with the cumulative probability of DAZL binding within clusters of proximal 3' UTR binding sites.

In our current study, we describe the computational identification of putatively new, and sometimes dual functions of RBPs in the regulation of mRNA translation. We integrate cross-linking immunoprecipitation (CLIP)-derived mRNA targets for 143 RBPs [12,33] with the transcriptomes and translatomes of 80 human hearts [34]—a tissue where translational control is known to play a central role in gene expression regulation [30,34,35]. We show that the expression levels of many, but not all, investigated RBPs indeed correlate with target mRNA abundance and/or translational efficiency *in vivo*. For a subset of 21 RBPs–including proteins with diverse previously described roles, such as the endoribonuclease G3BP1, the helicase DDX3X, the protease PUM1 and the deubiquitinase UCHL5 –we could independently assign dose-dependent effects to both mRNA levels and translational efficiencies of largely distinct sets of target genes, each involved in unrelated biological processes. Mechanistically, these target genes also appeared to be regulated independently, driven by differential affinity for protein-coding sequence length or 5' UTR structure.

Our results show that RBPs with more than a single role in human biology are likely to be more prevalent than currently anticipated. We postulate that multifunctional RBPs may use their functional plasticity in a condition- or compartment-specific manner to regulate gene expression at multiple levels, for separately defined sets of target genes.

## Results

### RNA-binding protein abundance determines the efficiency of target gene translation

With the aim to define which RBPs can influence more than a single stage of gene expression control, we first determined whether RBP abundance can have predictive value for the extent of target gene regulation in the human heart. To this extent, we compiled protein-RNA interactions for 143 cardiac-expressed RBPs, consisting of the muscle-specific RBM20 [33] and 142 mostly ubiquitously expressed RBPs previously characterized in depth as part of ENCODE [12] (see Methods, **S1A Fig** and **S1 Table**). We then correlated the expression levels of these 143 RBPs with mRNA abundance and translational efficiencies (TE) of 11,387 cardiac-expressed genes across 80 human hearts—the largest set of matched human tissue transcriptomes and translatomes that is currently available [34]. mRNA abundance was measured using RNA-seq counts normalized per gene and per sample, whereas TE was defined as the ratio of Ribo-seq and RNA-seq counts normalized for each individual gene and sample (see Methods). This revealed a clear quantitative relationship between RBP expression levels and the extent of target gene expression control (e.g., translation rates: **Figs 1A** and **S1B**).

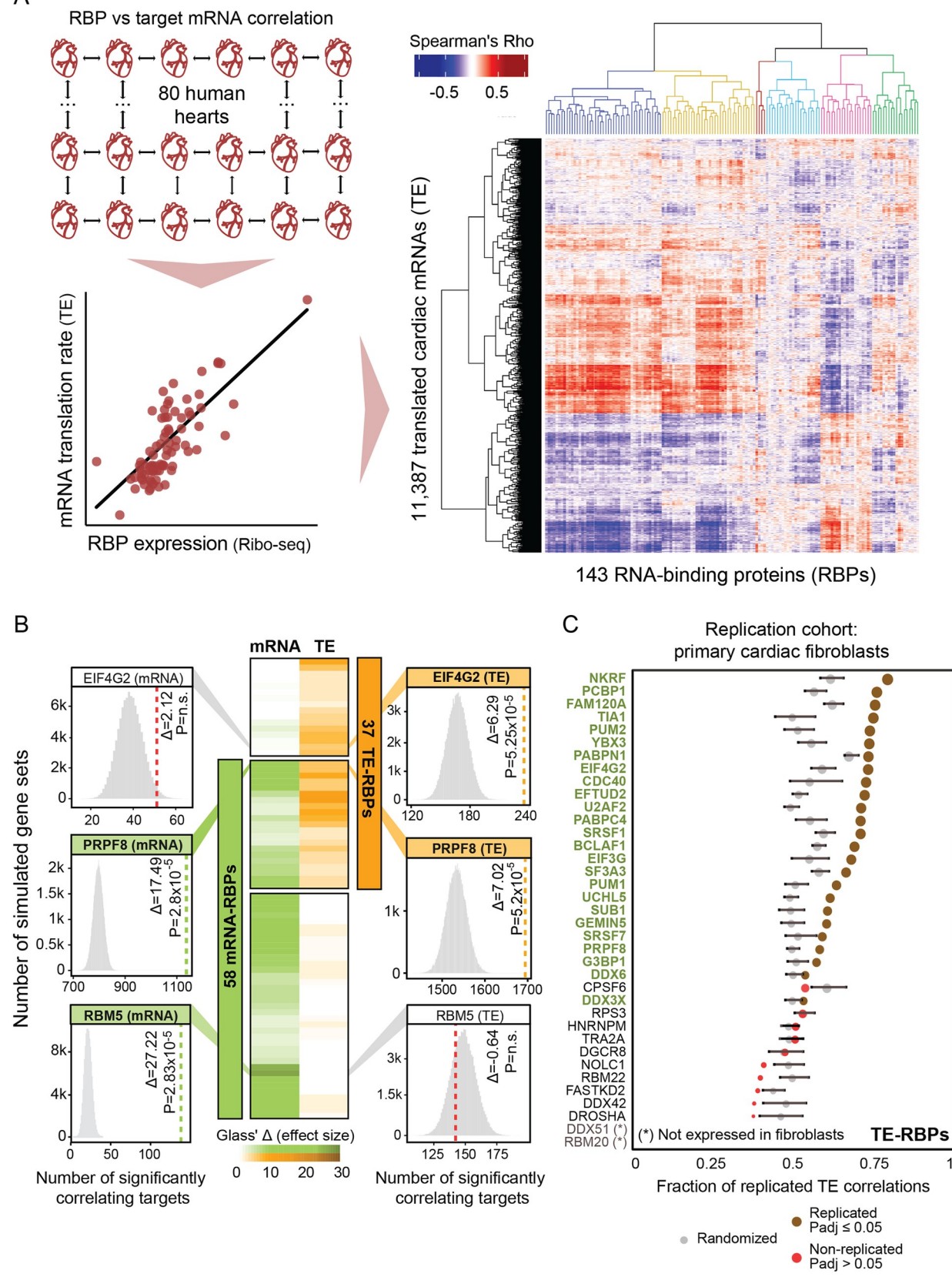

**Fig 1. RNA-binding protein abundance predicts target translational regulation.** **(A)** Schematic of the RBP-target correlation approach. Using the quantified Ribo-seq and RNA-seq data from 80 hearts, pairwise RBP versus target mRNA abundance or translational efficiency correlations were calculated. A heatmap with hierarchically clustered translational efficiency Spearman's Rho correlations of RBPs and translated mRNAs in the human heart are shown. Six clusters of coregulated RBPs are highlighted (See also **S1 Table**). **(B)** Heatmap with Glass' △ scores that quantify the effect size of the witnessed significance of associations between RBPs and target gene mRNA abundance and TE. Only significant RBPs are shown: 37 TE-RBPs (orange) and 58 mRNA-RBPs (green). For three selected RBPs (one per category), histograms illustrate the significance of the calculated associations. **(C)** Dot plot displaying the fraction of translational efficiency RBP-target correlations that can be replicated in an independent set of primary cardiac fibroblasts [30]. For each RBP, the significance of the replication was evaluated by comparing the replicated fraction between observed and randomized sets and it is represented as a brown (significant) or red (non-significant) dot. The size of the dots indicates the strength of significance (-log10 ($p_{adj}$)) and grey dots correspond to the fraction of replicated correlations in randomized sets. Error bars indicate mean values with standard deviation (SD).

Next, we calculated the frequency with which mRNA levels and translational efficiencies of the CLIP-derived target genes correlated significantly with the abundance of each RBP. For this, we statistically evaluated associations through the sampling of 100,000 equally sized sets of simulated theoretical targets out of the 11,387 translated genes in the human heart, yielding 58 RBPs with significant associations (empirical $p_{adj} \leq 0.05$) with target mRNA abundance (hereafter denoted as "mRNA-RBPs" that regulate "mRNA targets") and 37 RBPs with significant associations with target translational efficiencies ("TE-RBPs" regulating "TE targets") (**Figs 1B** and **S1C**). The effect size of each significant association was quantified using the Glass' Delta score (△) [36], a measure of the difference between the experimental and simulated groups divided by the standard deviation of the control. These significant targets, for instance, included the candidate tumor suppressor and ubiquitous splicing regulator RBM5 [37], which we identified as a cardiac mRNA-RBP influencing mRNA abundance of at least 138 correlating targets ($p_{adj} = 2.83 \times 10^{-5}$; Glass' Δ = 27.2). Reassuringly, our strategy validated known TE-RBPs such as the eukaryotic translation initiation factor EIF4G2, whose expression dynamics could be associated with target gene translational efficiencies of at least 235 correlating targets expressed in the human heart ($p_{adj} = 5.26 \times 10^{-5}$; Glass' Δ = 6.3) [38,39] (**Fig 1B**). Importantly, we could replicate our calculations for 25 out of 37 depicted TE-RBPs in an independent, though smaller cohort of primary cardiac fibroblast translatomes (n = 20; [30])—a system previously explored to identify RBPs with key roles in cardiac fibrosis (**Fig 1C**).

## Positive and negative control of translation by known and unknown factors

We detected 27 (out of 37) TE-RBPs without prior evidence of regulating mRNA translation, including 4 RBPs with no function assigned to their RNA binding ability at all (NKRF, FAM120A, SUB1 and UCHL5) (**S2A Fig**). To define how these RBPs interact with existing RBP networks to regulate target gene translation in a coordinated manner, we hierarchically clustered the correlation coefficients of all 37 TE-RBPs and their CLIP targets. This primarily divided the matrix in two distinct groups of TE-RBPs with marked opposite effects on target TE, indicative of competition and/or cooperation between subsets of RBPs (**Fig 2A**). Interestingly, the presented method joins known and novel TE-RBPs with opposing or concordant directionality of regulation on shared targets (**Fig 2B**). For instance, depending on the shared target gene bound by the splicing factor U2AF2 and the protease UCHL5, completely opposite effects on TE could be observed and independently replicated (**Figs 2B,** S2B and S2C). Moreover, most shared targets were not affected by RBP collinearity (**Fig 2B**, see Methods). Although these shared modes of target regulation were in part concordant with protein-protein interactions annotated in the STRING database [40], a subset of coregulatory "RBP hubs" contained proteins with previously unknown functional similarities (e.g., UCHL5 and U2AF2).

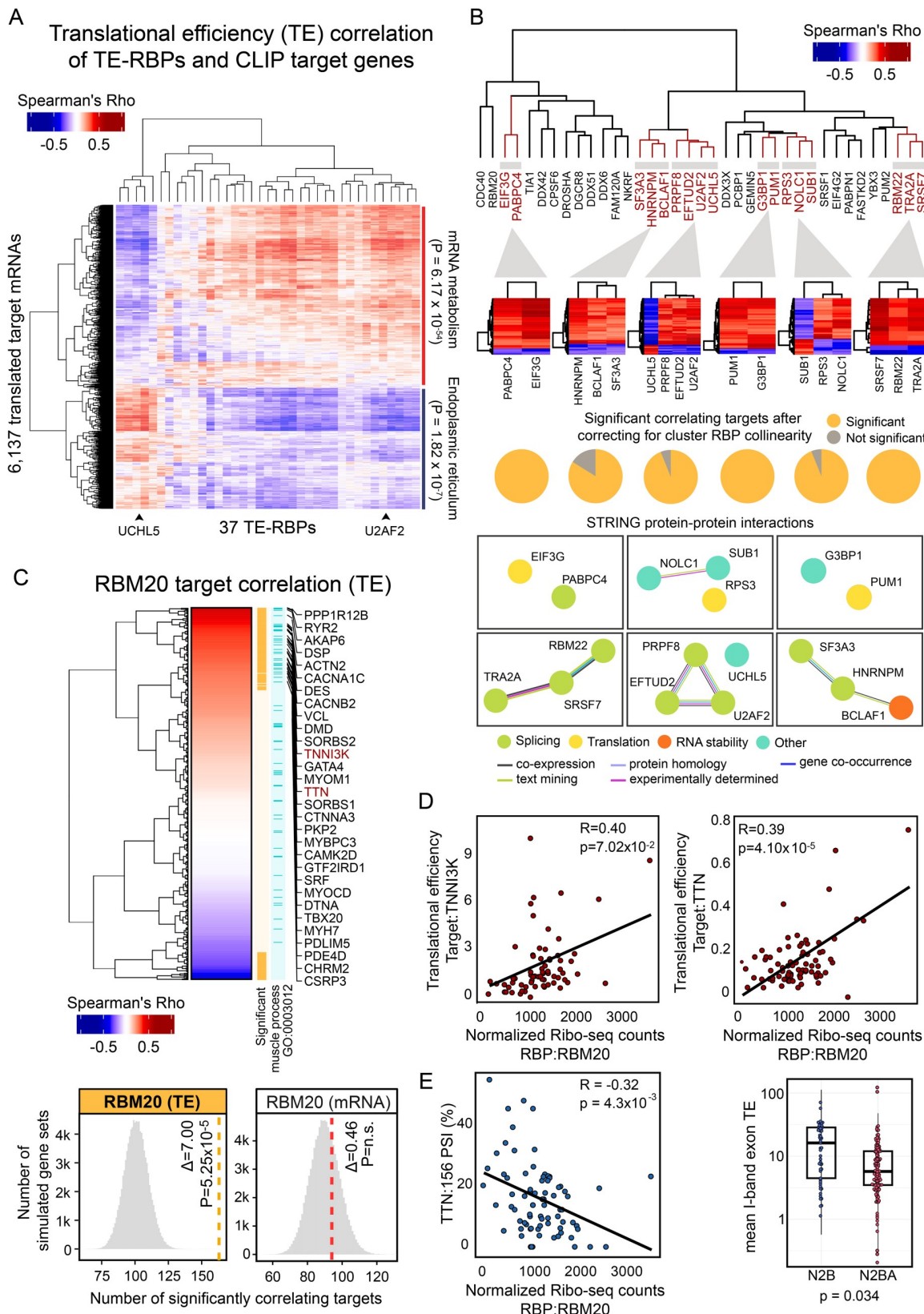

**Fig 2. CLIP analysis identifies coregulated *in vivo* targets of novel master regulators of translation in the human heart. (A)** Heatmap displaying the hierarchically clustered correlations between the cardiac expression levels of the 37 TE-RBPs (as determined by normalized Ribo-seq expression) and the cardiac TE of 6,153 correlating target genes. Each of the significantly correlating target genes was previously found to be bound by at least one of these 37 TE-RBPs based on CLIP experiments (see Methods). The clustering separates two groups with opposite effects on TE, whose targets are enriched for mRNA metabolism ($p_{adj}$ = 6.17 x $10^{-54}$) and endoplasmic reticulum ($p_{adj}$ = 1.82 x $10^{-7}$) GO terms, respectively. **(B)** Dendrogram with hierarchically clustered TE-RBPs based on pairwise RBP-RBP overlaps. Shared target genes of all paired RBPs were included for clustering. Bottom heatmaps with translational efficiency correlations of selected RBP clusters and shared significant targets. These plots illustrate distinct cooperative and competitive RBP-target regulation modes. Pie charts illustrate the fraction of targets that remain significant after correcting for RBP collinearity per cluster. STRING protein-protein interaction networks [40] from selected RBP clusters reveal functional association of coregulated RBPs. Colours in edges and nodes indicate the sources of STRING evidence and known RBP functions. **(C)** Heatmap with hierarchically clustered Spearman's Rho correlation scores of *RBM20* and the translational efficiency of the predicted target genes. Significant correlating targets (n = 163, $p_{adj} \leq 0.05$) and targets involved in muscle process (GO: 0003012) are highlighted in orange and light blue colours respectively. A list of sarcomere gene targets positively correlating with *RBM20* is displayed. Selected bottom histograms illustrate the significance of *RBM20* with correlating TE targets and the absence of significance with correlating mRNA targets. **(D)** Scatter plots representing the correlation between *RBM20* expression (as measured by normalized Ribo-seq counts; x-axis) and the translational efficiency (TE; y-axis) of two sarcomere genes: *TNNI3K* and *TTN*. Score and level of significance of the two Spearman's correlations are displayed. **(E)** Left: Scatter plot showing the correlation between normalized *RBM20* expression levels (as measured by Ribo-seq) and the percent spliced in (PSI) of *TTN* exon 156. Right: Box plot comparing average *TTN* I-Band isoform-specific TEs, showing a marked difference between *TTN* isoform N2B (ENST00000460472), displaying a significantly higher TE than *TTN* isoform N2BA (ENST00000591111) (Wilcoxon rank sum test, p-value = 0.034).

Amongst the 27 potentially new TE-RBPs was the muscle-specific and dilated cardiomyopathy-associated splicing regulator RBM20, whose expression correlated particularly well with the TE of 163 experimentally validated target genes (out of 561 total targets; Glass' Δ = 7.0; **Fig 2C**). Importantly, RBM20 levels specifically influenced TE and had no impact on overall mRNA abundance or stability. Most RBM20 targets, including the sarcomere genes *TTN* and *TNNI3K*, correlated positively (i.e., higher RBM20 expression associates with increased target gene TE; **Fig 2D**) and especially those positively correlating targets showed strong enrichment for muscle function processes (GO:0003012, $p_{adj} \leq 5.97 \times 10^{-16}$) (**Fig 2C**). To investigate a possible connection between RBM20-mediated mRNA splicing and the subsequent efficiency of mRNA translation, we evaluated whether splicing rates of known target exons correlated directly with TE. Splicing rates were quantified using calculated percent spliced in (PSI) scores, a metric that evaluates the efficiency of splicing for a specific exon into the transcript population of a gene (see Methods). For 66 out of 163 (± 40%) translationally regulated RBM20 target genes, the extent of alternative splicing indeed correlated with RBM20 abundance. A clear example is the exon inclusion measured across the *TTN* I-band, whose exons are only included in the longer *TTN* N2BA isoform. These I-band exons specifically drive the negative correlation of RBM20 expression with overall *TTN* TE, indicating that their inclusion reduces the efficiency with which *TTN* can be translated. We had previously observed that *TTN* translation rates are strongly isoform-dependent [34] and can now mechanistically connect this to splicing control by RBM20 (**Fig 2E**), a consequence that seems generalizable for more muscle-specific RBM20 targets, including other sarcomere components (**Fig 2D**). Although the precise mechanism through which RBM20 influences TE remains unknown, this RBP may omit the inclusion of exons with inefficient codon translation rates or exons that impact the stability or structure of the transcript, both of which influence protein synthesis rates [28,41].

## Multifunctional RBPs including DDX3X and G3BP1 regulate mRNA abundance and translational efficiency of independent sets of target genes

Among the 74 RBPs that correlated significantly with target gene TE (37 TE-RBPs) or mRNA abundance (58 mRNA-RBPs), a subset of 21 RBPs could be associated independently with both molecular traits (**Figs 1B** and **3A**). To investigate whether the association with mRNA abundance and TE was interrelated (and hence successive, i.e., higher target expression drives

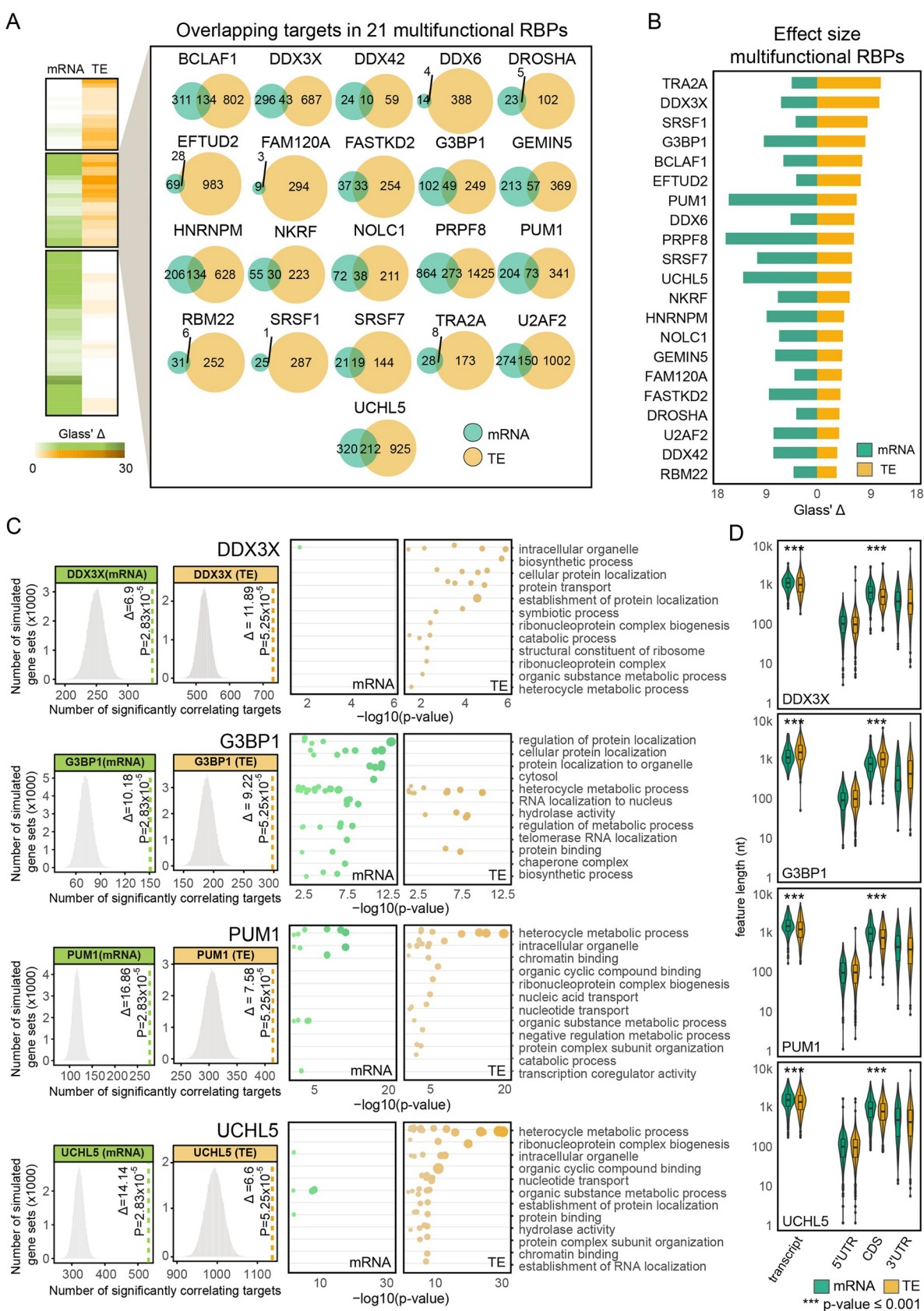

**Fig 3. Multifunctional RBPs regulate translation of distinct sets of target genes. (A)** Heatmap with Glass' △ scores quantifying the effect size of the witness effects for mRNA and TE correlations. Both effect sizes are significant for a highlighted set of 21 multifunctional RBPs. For this set of RBPs, individual Venn Diagrams representing the overlap in the total number of mRNA and TE targets are displayed**. (B)** Bar plot quantifying the magnitude of mRNA and TE effect size (Glass' △ scores) for multifunctional RBPs. RBP effect sizes are largely independent of the mode of regulation. **(C)** Selected histograms and dot plots illustrating the significance of RBP-target correlations and the enrichment of GO terms for the targets bound by 4 multifunctional RBPs: DDX3X, G3BP1, PUM1, and UCHL5. For each RBP, the 12 most significant parental GO terms are displayed. For three of the RBPs, mRNA and TE targets exhibit different enrichment of significant GO terms. **(D)** Box plots with transcript, 5' UTR, CDS, and 3' UTR sequence lengths in nucleotides for mRNA and TE targets corresponding to the four selected multifunctional RBPs in (C). A total of 9 multifunctional RBPs bind targets with significantly different CDS lengths (Wilcoxon rank sum test). See also **S3 Fig**.

increased TE), we examined the sets of genes that correlated significantly with either trait. This revealed a very limited overlap between correlating target genes for all 21 RBPs ($16.71 \pm 8.19\%$, **Fig 3A** and **S2 Table**). To substantiate this observation, we compared the trait-specific strength of the effect sizes (again as measured by the Glass' Delta ($\triangle$) [36]) and found largely no relation (**S2 Table**) between the correlations as independently witnessed for both traits, confirming the absence of a carryover effect (**Figs 3B** and **S3A**). This led us to denote these RBPs as "multifunctional RBPs"—context-specific RBPs whose functional outcome depends on the set of mRNAs it targets. A key example appears to be the multifunctional RBP DDX3X [42,43], whose abundance correlates significantly with the mRNA levels of 339 target genes ($p_{adj} = 2.83 \times 10^{-5}$; Glass' $\Delta = 6.9$) and the translational efficiency of 730 target genes ($p_{adj} = 5.25 \times 10^{-5}$; Glass' $\Delta = 11.89$), of which only 43 targets overlap between both sets (**Fig 3A** and **3C**). The consequences of DDX3X binding for mRNA abundance (positive correlation) or TE (negative correlation) are opposite, though this is not the case for all multifunctional RBPs (**S3B Fig**). Three other multifunctional RBPs similarly act as repressors of translation whilst having a positive effect on mRNA abundance (DDX6, NKRF, GEMIN5), one RBP shows the exact opposite behavior (FAM120A), and all others have concordant roles at both layers of control (e.g., TRA2A, FASTKD2, SRSF1).

Of note, the TE and mRNA target genes of multifunctional RBPs can be enriched for separate biological processes, indicating that duality can contribute to independent biological outcomes. For instance, correlating DDX3X and UCHL5 TE targets code for proteins involved in RNA splicing (GO:0008380, $p_{adj} = 7.70 \times 10^{-30}$), while their mRNA targets did not show any clear functional enrichment (**Fig 3C**). For G3BP1, mRNA targets code for proteins involved in localization to nuclear body (GO:1903405, $p_{adj} = 5.13 \times 10^{-12}$), whereas this is not the case for translationally regulated targets, which are enriched for RNA splicing (GO:0008380, $p_{adj} = 1.61 \times 10^{-10}$). Such biological discrepancies are not always present: independent of the mode of regulation, both types of PUM1 targets (TE or mRNA) appear to code for proteins involved in mRNA processing (GO:0006397, TE $p_{adj} = 2.64 \times 10^{-22}$, mRNA $p_{adj} = 6.40 \times 10^{-13}$).

## Differential affinity of multifunctional RBPs for CDS lengths and 5' UTR structures

Dual- or multiprotein functionality can be achieved through context-specific differences in subcellular localization [44], interaction partners [45,46], or the presence of multiple RNA-binding domains [47]—all of which can fine tune or restrict the subset of recognized target genes. Based on published immunofluorescence imaging-based evidence of subcellular RBP localization [12], 13 out of 21 multifunctional RBPs indeed localized equally well to both nucleus and cytosol, suggesting functionality in both compartments (**S3 Table**).

We additionally explored the relative position of CLIP binding sites in target genes (i.e., the position of binding within the mRNA: 5' or 3' UTR, CDS, or intronic). Most of the RBPs (including e.g., DDX3X) were recruited to target gene regions in different proportions, but we

observed no marked difference in binding positioning between mRNA and TE targets (**S3B Fig**). However, for SRSF7 and GEMIN5, mRNA and TE targets were enriched for intronic and CDS binding sites respectively. For DDX6, we observed an enrichment of TE targets with 3' UTR binding sites, of which 87% correlated negatively with the RBP. A recent study describing a tethered function assay to assess RBP functionality found a highly concordant role for DDX6 in the repression of translation, which resulted directly from its recruitment to target gene 3' UTRs [31]. This is in line with previously described repressive functions of DDX6 during mRNA translation [48].

For DDX3X and 10 other RBPs, we noticed a significant change in target transcript length, mostly explained by differences in target CDS length, which slightly increased or decreased between TE and mRNA targets (**Figs 3D** and **S3C**). The most significant changes in CDS length were seen for GEMIN5 (decrease for TE targets; 2,226 nt vs. 1,519 nt; $p_{adj}$ = $3.66 \times 10^{-9}$), PRPF8 (decrease for TE targets; 2,243 nt vs. 2,076 nt; $p_{adj}$ = $1.03 \times 10^{-8}$), DDX3X (decrease for TE targets; 1,659 nt vs 1,376 nt; $p_{adj}$ = $2.81 \times 10^{-7}$) and G3BP1 (increase for TE targets; 1,985 vs. 2,798 nt; $p_{adj}$ = $6.11 \times 10^{-5}$).

Our attention was drawn to the DEAD-box helicase DDX3X, which regulates translation initiation by interacting with, and subsequently disentangling, highly structured RNA sequences [49–52]. DDX3X binds 5' UTRs and the small ribosomal unit to facilitate the translation of a subset of mRNAs with long and structured leader sequences [52]. In order to define if additional RBPs may be required for, or involved in, translation initiation at targets with highly structured 5' UTRs, we looked into the 5' UTR's minimum free energy (MFE, length normalized) of TE and mRNA target genes, for each of the multifunctional RBPs. The MFE defines the most thermodynamically probable secondary structure for each RNA sequence, with lower MFE values pointing to more complex and structured predicted conformations. We observed that between the positively and negatively correlating target translational efficiencies of 17 out of 21 multifunctional RBPs, 5' UTR sequences differed in structural composition (**Fig 4A**). In contrast, there were poor to almost no differences for the significantly correlating mRNA targets of the same RBPs. Strikingly, three RBPs exhibited by far the strongest MFE differences between positively and negatively correlating targets: next to DDX3X ($p_{adj}$ = $9.47 \times 10^{-47}$), those were the core spliceosome factors PRPF8 ($p_{adj}$ = $2.70 \times 10^{-29}$) and EFTUD2 ($p_{adj}$ = $1.69 \times 10^{-30}$) (**Fig 4A and 4B**). Positively and negatively correlating targets also displayed minor differences in 5' UTR lengths (**S4A Fig**), but the difference in MFE normalized by length was much greater than the small difference observed in UTR length, which was therefore corrected for. Moreover, the effect of MFE was still highly significant after controlling for UTR length (**S4B Fig**). The targets shared between these RBPs displayed similar directions of correlation with the three RBPs (**Fig 4C**) suggesting that these three RBPs might rely on a similar mechanism for translational regulation that is dependent on the 5' UTR structures of shared targets. Interestingly, we found that positively and negatively correlating targets were involved in different functions and cellular compartments (**S4C Fig**). For instance, positively correlating targets from DDX3X and EFTUD2 were enriched for proteins that constitute ribonucleoprotein complexes, in line with a recent study that showed strong translational downregulation of ribonucleoproteins after knockdown of DDX3X [53]. On the contrary, negatively correlating targets of DDX3X and EFTUD2 were enriched for proteins localizing to membrane-bounded cytoplasmic organelles, such as the mitochondria.

## Discussion

Increasing evidence suggests that RBPs can act as multifunctional gene expression regulators [14,54]. Here, we built an *in-silico* method for the large-scale analysis of RBP-driven regulation

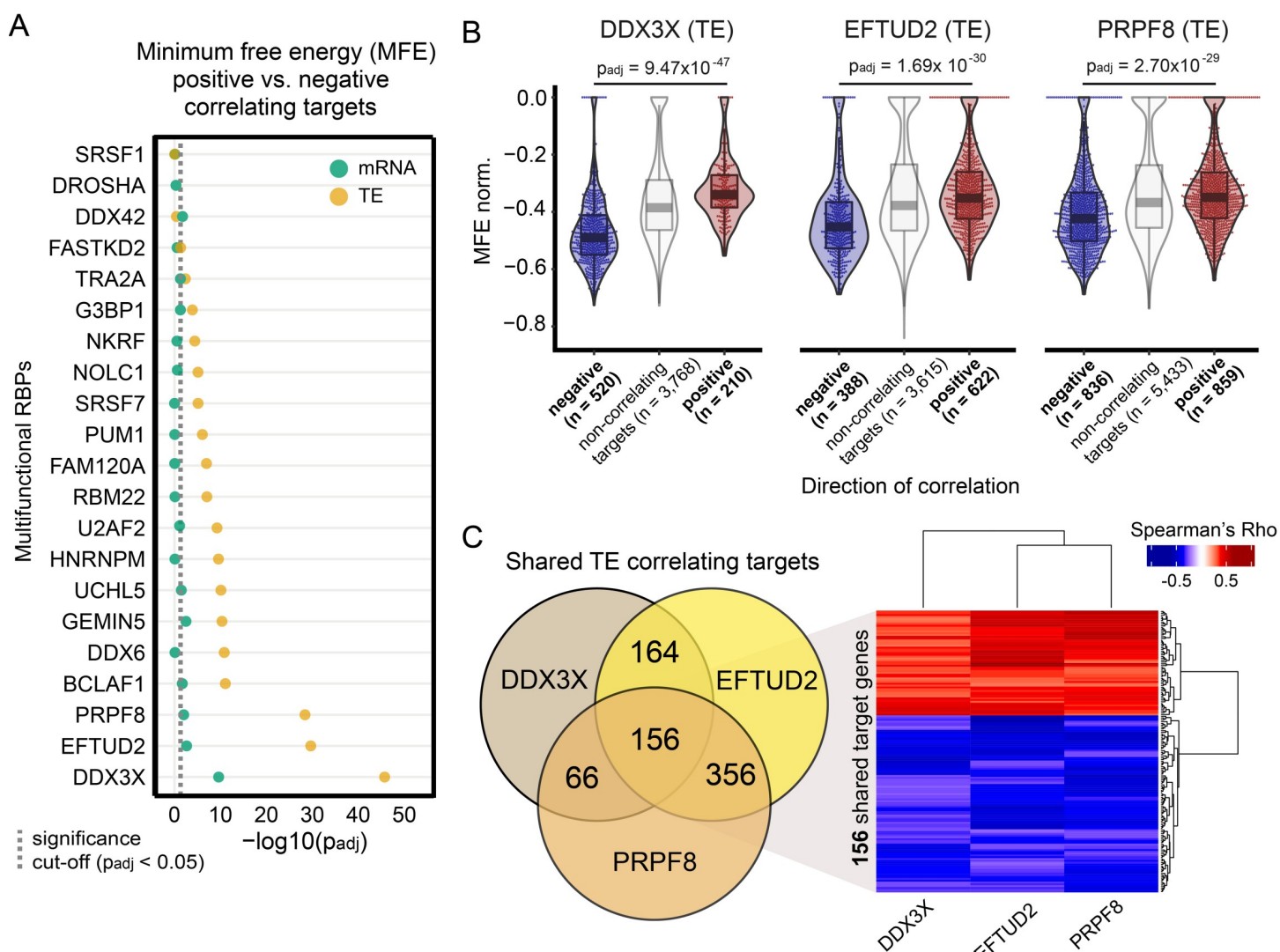

**Fig 4. Differential affinity of multifunctional RBPs for 5' UTR structures often drives opposite quantitative TE effects. (A)** Dot plot displaying the significance of the differences in 5' UTR minimum free energy (MFE, normalized by length) between target genes that correlate positively or negatively with each multifunctional RBP. Significance values are calculated separately for mRNA (green) and TE (brown) targets. Adjusted p-values are shown on a -log10 scale and calculated using the Wilcoxon Rank Sum test and only 5' UTR sequences with a minimum length of 20 nucleotides were evaluated. A dashed vertical line indicates the minimum adjusted p-value to consider the differences in MFE as significant ($p_{adj} < 0.05$). **(B)** Box and violin plots with length normalized MFE scores for positively and negatively correlated TE targets corresponding to the three selected multifunctional RBPs with the highest significance (Wilcoxon rank sum test) in **Fig 4A** (DDX3X, EFTUD2, PRPF8). For comparison, non-correlating target genes were included in the panel figure. **(C)** Three-way Venn Diagram representing the overlap in the number of TE targets for the three selected RBPs. The heatmap represents TE correlations of 156 shared target genes for the three cases.

using correlation as a proxy for mRNA abundance and translational efficiency (TE) of target genes across 80 human heart samples. Our approach underscores the functional importance of RBP expression fluctuations in the control of gene expression, a mechanistic feature recently highlighted by others *in vitro* and *in vivo* to assign new unknown roles for RBPs [30–32]. We exploited the quantitative effect of RBPs on known target genes to implicate 74 RBPs in the regulation of mRNA abundance or translation.

We discovered 27 RBPs with previously unknown roles in translation, some of which have well-characterized functions in other biological processes, including mRNA splicing. Previous work revealed a handful of splicing factors that can independently mediate post-splicing

activities, such as mRNA translation [55–57]. Instead, alternative splicing mediated by splicing factors can cause downstream effects on the translation of spliced target genes, depending on the qualitative decision of which mRNA isoforms are being produced [19–29]. Remarkably, the high fraction of splicing factors that we find to influence translation would suggest previously unanticipated roles for many more splicing regulators in this process. A prominent example is the muscle-specific and cardiac disease-relevant splicing regulator RBM20. RBM20 is a splicing repressor that modulates isoform abundances of many sarcomere genes [33,58]. Here we demonstrate that RBM20 expression correlates positively with the TE of *TTN* and other sarcomere genes, suggesting that nuclear splicing control can impact cytoplasmic protein synthesis. While we had previously reported that *TTN* isoform-specific changes in TE exist [34], we now for the first time show that RBM20 has a crucial impact on *TTN*'s translational output. Future functional studies should determine the specific mode of action of the splicing factors involved in translation and whether any isoform-specific characteristics contribute to the observed differences in TE.

Besides the discovery of potential novel functions for a subset of RBPs, we provide evidence that 21 RBPs can modulate both target mRNA abundance and TE—a class of RBPs that we classify as "multifunctional RBPs". Multifunctional RBPs appear to be involved in the regulation of mRNA abundance and TE of distinct groups of target genes. These target genes can be concordantly or discordantly regulated on either layer of gene expression control. We discovered that the specific affinity of several RBPs to structural properties of mRNAs, such as protein-coding sequence length, UTR length or RNA secondary structure, contribute separately to the observed independent effects on mRNA abundance or translation. Hence, further experimental validation is required to confirm the quantitative effect of each individual multifunctional RBP and identify the precise mechanisms behind this dual functionality. In support of our findings, a recent study [32] has inspected the connection between the binding kinetics of the RBP DAZL and its effect on mRNA abundance and translation of specific sets of target genes, identifying several 3' UTR features–UTR length, presence of binding clusters, distance to the polyadenylation site–that are linked to the trait-specific regulation of different groups of targets. In addition, the usage of different ribosome binding domains, the recognition of alternative RBP motifs and the presence of binding sites located in different gene regions (i.e., UTRs, CDS, or introns) can also be indicative of RBP multifunctionality [59]. Nevertheless, we found these mRNA characteristics to remain largely unchanged for the targets of most multifunctional RBPs identified in this study.

For a subgroup of multifunctional RBPs, we noticed that the targets regulated on the transcriptional or translational level represent functionally different gene classes. Indeed, the observed biological diversity in our study seems to match the condition-specific regulatory complexity that needs to be achieved by a single RBP. For instance, this appears to be the case for G3BP1—a known multifunctional RBP that can selectively compartmentalize specific sets of mRNAs to stress granules, in order to reprogram mRNA translation under certain global stress conditions [60–62], as well as tissue-specific contexts such as atrial fibrillation in heart [63]. Additionally, G3BP1 plays an important role in DNA/RNA unwinding [64] and binds to specific RNA stem-loop structures to trigger mRNA degradation [65], which is essential for maternal mRNA clearance [66]. Another example is DDX3X, a DEAD box helicase which can respond to stressors (e.g., viral infections [67]) by switching subcellular compartments. DDX3X is involved in multiple processes required for RNA metabolism [42,43], for which it uses its capacity to unwind complex and structured 5' UTRs to promote translation initiation at selected subsets of mRNAs [43,49,52,68]. However, there is ongoing debate as to the precise roles of DDX3X and the mechanisms through which it regulates RNA metabolism [68], as it can act both as a repressor or activator of translation [69].

Our work points to an intricate relation between the direction of translation regulation and target 5' UTR structure, with the TE of certain targets being positively or negatively influenced by multifunctional RBP binding depending on the complexity of target 5' UTR sequences. Unexpectedly, our results show that increased levels of DDX3X correlate with a lower TE for targets with highly structured 5' UTRs. This seems to contradict recent *in vitro* reports where DDX3X knockdown in human cells [52] resulted in translational repression of mRNAs with structured 5' UTRs. Next to DDX3X, the strongest impact of the 5' UTR on translational output is observed for EFTUD2 and PRPF8, which display patterns of regulation highly similar to DDX3X, suggesting an analogous mode of action in the control of target translation rates. Surprisingly, EFTUD2 and PRPF8 are splicing factors which are part of the central component of the U5 snRNP spliceosome [70] and had not been implicated in translation before. However, the conserved GTPase EFTUD2 has sequence similarity to the translation elongation factor EF-2 [71], possibly explaining its capacity to influence translation. Both ancient paralogs may have evolved and diversified to complement each other.

Whereas dual functionality of extensively studied and characterized RBPs such as DDX3X and G3BP1 had (to a certain extent) been described previously, for a selection of other RBPs our results provide initial observations of dual functionality. For instance, UCHL5 (also known as UCH37) is a protease with RNA binding capacity that may be part of the INO80 chromatin remodeling complex [72], though its role within RNA metabolism is yet unknown. We established a quantitative relationship between UCHL5 expression and variability in mRNA abundance of genes involved in chromatin organization, as well as with changes in TE of genes involved in RNA splicing. Although UCHL5 shared target genes with core splicing factors (U2AF2, EFTUD2 and PRPF8), its effect on the TE of targets shared with these three splicing factors is completely opposite, suggesting contrasting regulatory behavior, and possibly competition.

Very little is known about the molecular processes that control RBP multifunctionality, although some possible mechanisms have been recently investigated, including the formation of heterogeneous RBP complexes [73,74], switches from monomers to multimers in a concentration-dependent manner [75], and changes in subcellular localization [55,76]. In our current study, the potential mechanisms behind the observed multifunctionality could not be explained in a uniform way: there appears to be no 'one size fits all' scenario. It is very probable that RBP multifunctionality is achieved by specific combinations of individual RBP and target features, whose precise dissection requires experimental follow-up into each individual multifunctional RBP. An RBP may bind distinct sets of RNA within the nucleus, though, for a subset of targets, the consequences of binding may only become apparent at a later stage of gene expression (e.g., a change in transcript isoform production that is accompanied by a downstream effect on TE). Alternatively, multifunctional RBPs may physically take part in multiple stages of gene expression by adapting subcellular localizations. For instance, HNRNPM (one of the core splicing ribonucleoproteins that we found to influence both target gene mRNA abundance and TE) localizes to nucleus [16] but can be exported to the cytosol to induce cap-independent translation upon hypoxia [55]. Another example, DDX3X, shuttles between nucleus and cytosol [67]. It remains to be established if RBPs with shared targets bind these targets simultaneously, or if there is sequential crosstalk of RBPs and other proteins in the control of target expression. RBP abundance may also respond to target availability and not *vice versa*, possibly explaining why we find many splicing regulators to rank highly within the RBP hierarchy.

There are limitations to the methods we employ to define biological roles for RBPs and assign targets to each. First, we note that significant associations also occur in the randomly sampled target sets, which indicates a certain level of fuzziness in RBP-target assignment. This

appears inevitable for RBPs with general functions in expression control, which bind and share large numbers of targets. However, together with the calculated Glass' delta, the distributions of significant correlations in the 100,000 randomly sampled target sets give us a good indication of our false discovery rate in target assignment. Particularly for investigations into the effects of RBP binding on individual target genes, the presence of potential false-positive significant correlations indicates that independent experimental validation is warranted. A second limitation includes the use of eCLIP experiments from two cell lines unrelated to the heart. Our results would benefit from large-scale human heart-specific CLIP experiments that better complement the tissue-specificity of the cardiac expressed genes. Nevertheless, all discovered multifunctional RBPs are ubiquitously expressed across human tissues (**S1A Fig**), suggesting that these proteins are major players in RNA transcriptional and translational regulation, contributing to both global and cell-line or tissue-specific manifestations.

In conclusion, our results illustrate unanticipated complexity in RBP-RNA interactions at multiple consecutive levels of gene expression. This warrants future in-depth experimental research into the identified RBPs in human cardiac biology. Understanding how RBPs cooperate, communicate, interact, and compete across subcellular compartments and in response to changing conditions will be essential to fully comprehend the quantitative nature of the regulatory principles that underlie mRNA metabolism.

## Methods

### Ribosome profiling and RNA sequencing data analysis

We re-analyzed ribosome profiling (Ribo-seq) and matched RNA-seq datasets from 80 human hearts that we generated and published previously (EGA accession code: EGAS00001003263) [34]. In short, Ribo-seq reads were clipped for residual adapters using FASTX toolkit [77]. Reads mapping to the mitochondrial RNA, ribosomal RNA and tRNA sequences were removed from downstream analysis. Full length paired mRNA-seq reads ($2 \times 101$nt) were trimmed to 29-mers (average length of Ribo-seq reads) to establish a comparable analysis of both Ribo-seq and mRNA-seq datasets and avoid any mapping or quantification bias due to different read length or filtering. Next, Ribo-seq and trimmed mRNA-seq reads were mapped to the human reference genome (GRCh38, Ensembl v87) using STAR v2.5.2b [78] with maximum of 2 mismatches and -seedSearchStartLmaxOverLread = 0.5. Quantification of gene expression was performed by counting reads mapping to coding sequence (CDS) regions of annotated protein-coding genes, using HTSeq v0.9.1 [79]. Gene counts were normalized by estimating the size factors simultaneously on Ribo-seq and RNA-seq datasets using DESeq2 v1.12.4 [80]. This joint normalization is required to compare both measures of gene expression [81]. Translational efficiency (TE) was calculated on the Ribo-seq against RNA-seq ratio for each individual gene and sample, as described previously [34].

### Identification of RBP targets from published eCLIP and HITS-CLIP data

Processed eCLIP data of 150 RBPs were obtained from ENCODE (30) for HepG2 (n = 103) and K562 (n = 120) cell lines. Datasets consisted of BED files containing eCLIP peaks and BAM files containing reads mapped to the human genome (GRCh38.p10/hg38). The identification of robust eCLIP peaks across replicates and cell lines was performed as suggested by Van Nostrand and colleagues [12]. First, we used BEDTools [82] to quantify the coverage of each predicted peak using input (mock) and immunoprecipitation (IP, antibody against RBP) BAM files. Next, for each peak, the relative information content was defined as $p_i \times \log2 (p_i/q_i)$, where p and q are the sum of reads mapping to the peak in IP and negative control respectively. The information content was used to calculate the Irreproducible Discovery Rate (IDR)

[83], a parameter indicating reproducible peaks across biological replicates. A significant and reproducible peak was defined meeting an IDR cut-off $< 0.01$, p-value $\leq 10^{-5}$ and fold-enrichment (FC) $> 8$. In case two or more peaks overlapped the same genomic region, the most significant one was included in the peak table. Additionally, non-overlapping peaks were pooled into a single table, in order to get a complete set in both cell lines. While CLIP data was produced in a non-cardiac setting, CLIP signals are usually preserved among similarly expressed genes of the same RBP independent of the cell line, with peak differences instead reflecting cell type-specific expression rather than binding affinity [12]. Additionally, for the muscle-specific splicing repressor RBM20, which was not part of the ENCODE dataset but included for its importance for cardiac splicing and heart disease [33,58], significant rat RBM20 HITS-CLIP targets were obtained from Maatz et al. [33] and converted to GRCh38.p10/hg38 genomic coordinates. Only 143 RBPs with expression in human heart tissue were kept (mean FPKM across samples $>1$; 142 ENCODE RBPs and RBM20).

Overall, we retrieved an average of 4,300 eCLIP-seq peaks per experiment. Finally, we mapped these peaks to the annotated transcriptome (Ensembl v.87) and, for each RBP experiment, all the genes supported by at least one CLIP-seq peak were defined as putative target genes.

## RBP-target correlation and clustering

For RBP-target correlations and clustering we included genes expressed in the human heart (mean FPKM across samples $> 1$) with at least one Ribo-seq and mRNA-seq read in a minimum of 20 samples (n = 11,387). Next, for pairwise complete observations, we calculated Spearman correlations between the expression level of the RBP (as measured by Ribo-seq) and either target gene mRNA-seq counts or translational efficiency. Only target genes that showed a significant ($p_{adj} \leq 0.05$) correlation after correction for multiple testing using the Benjamini-Hochberg approach [84] were retained for downstream analyses. The computed RBP-target correlation matrix was used to calculate the Euclidean distance followed by hierarchical clustering, to group RBPs with similar consequences on their target genes. Cluster visualization was done using heatmap.3 (https://github.com/obigriffith/biostar-tutorials/tree/master/Heatmaps). To ensure significant correlation between RBP and its target genes and exclude numerical relationship (collinearity) between two RBPs, we selected clusters of RBPs and target genes with similar expression profiles and calculated partial correlations. Statistical comparison was enabled by Fisher Z-transformation of two correlation coefficients to a normal distribution.

## Target gene enrichment

To identify RBPs that are putative modulators of target gene mRNA abundance and/or TE, we calculated the frequency with which target genes supported with CLIP-seq data correlated significantly with each RBP. We leveraged the significance of these correlating associations by generating 100,000 equally sized sets of theoretical targets out of all translated genes in the human heart; an approach that has previously been shown to be highly effective [30]. For each set, we quantified the amount of significantly correlating genes and compared the theoretical distribution against the actual observation applying an empirical test: *Empirical p-value = sum (theoretical targets ≥ true RBP targets) /100,000.*

Empirical p-values were corrected for multiple testing (Benjamini-Hochberg method). RBPs that showed a significant ($p_{adj} \leq 0.05$) enrichment of correlating CLIP-derived target genes were considered as putative regulators of mRNA abundance (n = 58) and/or TE (n = 37). It should be noted that, because of the fixed number of generated random sets, the

minimum empirical p-value that can be calculated after correction for multiple testing is 5.25 x $10^{-5}$. Hence, the empirical test cannot quantify the strength of significance for a specific observation. Instead, we calculated Glass' Delta ($\triangle$)[36] as a measure of the effect size, which is defined as the difference between the two target sets divided by the standard deviation of the theoretical group. *Effect size = (true RBP targets—mean(theoretical targets)) / sd(theoretical targets)*

## RBP expression across GTEx tissues

To determine the patterns of expression of each RBP across human tissues, we obtained expression data from the Genotype-Tissue Expression (GTEx) Project [85], a database that comprises a large set of samples corresponding to 54 human tissues. We used these data to determine the number of tissues with detectable (average TPM $\geq$ 1) or high (average TPM $\geq$ 10) expression of a given RBP (**S1A Fig**). An RBP was categorized as ubiquitously expressed if expression was detected in more than 30 tissues with a TPM $\geq$ 10.

## Replication of target regulation using a public fibroblast cohort

We retrieved raw RNA-seq and Ribo-seq data from a cohort of 20 primary cardiac fibroblast cultures stimulated with TGF-beta [30] and used it as a replication cohort. Raw data are available via the gene expression omnibus (GEO submission: GSE131112, GSE123018, GSE131111) repository. Read pre-processing, mapping, gene quantification and correlation analysis were done following the same procedures described above for the heart datasets (see 'Read mapping and gene quantification' and 'RBP-target correlation and clustering' subsections). To prove that the regulatory effect of RBPs in target translational regulation can be replicated in an independent dataset, we quantified the fraction of RBP-target correlations with similar direction of regulation in both fibroblast and human heart cohorts. Statistical significance of the observed replications was evaluated by running 10,000 permutations of the correlation coefficients in fibroblasts and comparing the fraction of shared directionality between both cohorts in observed and randomized sets.

## Analysis of differential exon splicing

To evaluate whether RBM20 could influence the TE of target genes by modulating isoform production ratios (exon in- or exclusion), we estimated exon splicing rates by calculating the percentage spliced in (PSI) for all exons of known and correlating RBM20 target genes, as described previously [86]. For PSI calculation, we re-mapped the 80 paired-end cardiac mRNA-seq (2 × 101nt) datasets to improve splice site coverage using STAR v2.5.2b [78], allowing a maximum of 6 mismatches.

## Functional analysis of RBP associations and target genes

Known and predicted RBP-RBP interactions were retrieved from the STRING database [40] with confidence network edges and default settings. Moreover, we assigned biological functions to define gene targets with gProfiler2 v0.1.9 (archive revision fof4439, [87]) and extracted enriched sets of 'child' and 'parent' GO terms for the individual sets mRNA and TE targets ($p_{adj} \leq$ 0.05). Significant GO terms that involve a minimum of 20 and a maximum of 500 genes were considered, to avoid the inclusion of too general terms that show significance due to beneficial input to term size ratios.

### Analysis of minimum free energy in 5' UTRs

We predicted 5' UTR secondary structures through energy minimization using RNAfold from the Vienna Package v2.4 [88]. Using the 5' UTR sequence of each target gene as input, minimum free energies (MFE) were calculated and length-normalized to observe differences in UTR complexity for target genes that are positively or negatively correlating with RBPs.

### General remarks on statistical analysis

Statistical analysis and generation of figures was done using R v3.6.2 [89]. A full list of tools and methods used for data analysis is stated in each corresponding Methods section. Statistical parameters such as *n*, median/mean, standard deviation (SD) and significance are named in the figures and/or the figure legends. The "*n*" represents the number of RBPs in **Figs 4, S3A** and S4.

## Supporting information

**S1 Table. Analysis information for 143 RBPs.** Table with all 143 cardiac expressed RBPs, number of mRNA and TE correlating targets and significance of correlations, clusters of coregulated RBPs, and average RNA expression levels in human left ventricle.
(XLSX)

**S2 Table. Multifunctional RBPs.** Table with all 21 multifunctional RBPs, number of target genes per molecular trait, and names and significance of the best 5 GO enrichment results.
(XLSX)

**S3 Table. Multifunctional RBPs' localization.** Table with all 21 multifunctional RBPs and their cellular localization (0: absent; 1: present).
(XLSX)

**S1 Data. RData object containing DESeq2 normalized poly(A) RNA-seq and Ribo-seq count matrices together, TE matrix (Ribo-seq/RNA-seq) and PSI count matrix.** To access this file, download R and load the RData object with the R function load: load(RData).
(ZIP)

**S2 Data. RData object containing correlation matrices, global result- and effect-size tables for all mRNA-RBPs and TE-RBPs as well as a table with MFE values for all target transcripts for each TE-RBP.** To access this file, download R and load the RData object with the R function *load*: load(RData).
(ZIP)

**S1 Fig. RNA-binding protein abundance predicts target translational regulation. (A)** Bar plot displaying the patterns of expression of the 143 RBPs across tissues. Average expression values in transcript per million (TPM) units were retrieved from the Genotype-Tissue Expression (GTEx) Project. Most of the RBPs are ubiquitously expressed across human tissues. **(B)** STRING protein-protein association networks from six coregulated RBP clusters (see also **Fig 1A**). Most of the clustered RBPs are involved in known functional interactions. **(C)** Heatmaps with Glass' △ scores for all 37 TE-RBPs quantifying the effect size of the witness effects for significant TE correlations.
(TIF)

**S2 Fig. CLIP analysis identifies coregulated *in vivo* targets of novel master regulators of translation in the human heart. (A)** Described functions by Van Nostrand et al. for the set of TE-RBPs. Functions related to translation (translation regulation and ribosome basic

translation) are highlighted with dark red boxes. **(B-C)** Scatter plots representing the correlation of heart **(B)** and primary cardiac fibroblasts **(C)** translational efficiencies between UCHL5 and U2AF2 and two shared targets, *KPNA4* and *MYL6*. UCHL5 and U2AF2 have marked opposite effects on their shared targets, indicative of a competitive effect replicated in two independent datasets. Scores and level of significance of the two Spearman's correlations are displayed.
(TIF)

**S3 Fig. Multifunctional RBPs regulate translation of distinct sets of target genes. (A)** Network representing multifunctional RBP-target interactions for both mRNA-RBPs (green) and TE-RBPs (brown) of strong correlating pairs. Blue lines indicate shared targets in both mRNA abundance and TE regulation of the same RBP. **(B)** Left: heatmap representing the average mRNA and TE RBP-target correlation values for all 21 multifunctional RBPs. Middle: heatmap representing differences in the relative proportion of feature binding sites (TE-mRNA) for all 21 multifunctional RBPs. Right: bar plot showing the overall proportion of feature binding sites for all 21 multifunctional RBPs. **(C)** Box plots with 5' UTR, CDS, and 3' UTR sequence lengths in nucleotides for mRNA and TE targets corresponding to the set of 21 multifunctional RBPs. For each target gene, the most abundant isoform is represented.
(TIF)

**S4 Fig. Differential affinity of multifunctional RBPs for 5' UTR structures often drives opposite quantitative TE effects. (A)** Box and violin plots with 5' UTR lengths for positively and negatively correlated TE targets corresponding to DDX3X, EFTUD2, and PRPF8. **(B)** Box and violin plots with length normalized MFE scores for positively and negatively correlated TE targets. We subsampled sets of 50 genes per group and RBP, so each of the groups had a similar distribution of 5' UTR lengths. For comparison, non-correlating target genes were included in the panel figure. **(C)** Enriched GO terms in the sets of positive and negative correlating targets for DDX3X, EFTUD2, and PRPF8. For each RBP, the 5 most significant GO terms are displayed.
(TIF)

## Acknowledgments

We would like to thank Prof. Dr. Markus Landthaler for his helpful feedback and discussions.

## Author Contributions

**Conceptualization:** Valentin Schneider-Lunitz, Norbert Hubner, Sebastiaan van Heesch.

**Formal analysis:** Valentin Schneider-Lunitz.

**Funding acquisition:** Norbert Hubner.

**Investigation:** Valentin Schneider-Lunitz, Jorge Ruiz-Orera, Sebastiaan van Heesch.

**Methodology:** Valentin Schneider-Lunitz, Jorge Ruiz-Orera, Sebastiaan van Heesch.

**Software:** Valentin Schneider-Lunitz, Jorge Ruiz-Orera.

**Supervision:** Norbert Hubner, Sebastiaan van Heesch.

**Visualization:** Valentin Schneider-Lunitz, Jorge Ruiz-Orera, Sebastiaan van Heesch.

**Writing – original draft:** Valentin Schneider-Lunitz, Jorge Ruiz-Orera, Sebastiaan van Heesch.

**Writing – review & editing:** Valentin Schneider-Lunitz, Jorge Ruiz-Orera, Norbert Hubner, Sebastiaan van Heesch.

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
