## [Decision Letter · Decision Letter 0]

28 Sep 2021

Dear van Heesch,

Thank you very much for submitting your manuscript "Multifunctional RNA-binding proteins influence mRNA abundance and translational efficiency of distinct sets of target genes" for consideration at PLOS Computational Biology. As with all papers reviewed by the journal, your manuscript was reviewed by members of the editorial board and by several independent reviewers. The reviewers appreciated the attention to an important topic. Based on the reviews, we are likely to accept this manuscript for publication, providing that you modify the manuscript according to the review recommendations.

Sincerely,

Greg Tucker-Kellogg, PhD

Associate Editor

PLOS Computational Biology

Daniel Beard

Deputy Editor

PLOS Computational Biology

[LINK]

Reviewer's Responses to Questions

**Comments to the Authors:**

Reviewer #1: Attached

Reviewer #2: Multifunctional RNA-binding proteins influence mRNA abundance and translational efficiency of distinct sets of target genes

by Schneider-Lunitz et al.

Schneider-Lunitz et al. combine the transcriptomes and translatomes of human hearts with the target preferences of human RBPs to deduce potential associations and identify new multi-functional regulators of mRNA stability and translation. Genome-wide measurements for mRNA abundance and translational efficiency (derived from matched Ribo-seq and RNA-seq) for 80 human hearts were taken from the authors’ previous study (Van Heesch et al, Cell, 2019). RBP targets were extracted from publicly available eCLIP data for 142 RBPs, plus RBM20 from an independent dataset. Using correlation analysis (Spearman), the authors identify significant associations between the expression of a given RBP and the mRNA abundance (58 RBP-mRNA associations) or translation efficiency (37 RBP-TE associations) of its targets. They find that many RBPs show shared modes of regulation, having overlapping targets with opposing or concordant directionality of regulation. Moreover, they identify 21 RBPs as multifunctional RBPs, which influence mRNA abundance and translational efficiency for different subsets of target RNAs.

Overall, this is an interesting and very well performed study. The manuscript is well written and the figures are of high quality. The analyses are based on solid data and statistical approaches. Specific points that would enhance the impact of the study are listed below.

Based on the high similarity in the regulatory patterns of some RBPs (Figure 2AB), the question arises whether such results could be influenced by colinearity in the RBPs’ expression and/or a certain fuzziness in the assignment of the target genes for some RBPs. The impact of colinearity could for instance be assessed by partial correlations for pairs of similar RBPs to test whether RBP1 still shows a significant association with its target genes when correlations are corrected for RBP2. Regarding the second factor, the validity of target gene assignment for a given RBP could be assessed by cross-testing target genes, i.e., correlating the target genes of RBP1 against RBP2 and vice versa.

Minor comments:

Figure 2A: How many and which genes went into this analysis? Is this the union of target genes of all RBPs or some selection?

“For instance, depending on the shared target gene bound by the splicing factor U2AF2 and the protease UCHL5, completely opposite effects on TE could be observed and independently replicated (Figure 2C, Figure S2B and S2C).”

Reference to Figure 2C seems wrong.

Figure 2D/E: Extend legends to clarify what units were used for the scatter plots

Figure S3B: This figure requires more explanation. What exactly is shown in the middle panel? It seems that some RBPs show a region-dependent effect. For instance, as expected, SRSF7 seems to tend to regulate mRNA abundance via intronic binding and translational efficiency via CDS binding.

Figure 4B: MFE values have been normalised by length, but it is unclear whether this really works. The authors should check for differences in lengths between categories and use length-matched subsamples if needed. Also, what is the MFE distribution for non-correlated genes for comparison?

Reviewer #3: This manuscript presents a bioinformatics study investigating 143 RNA-binding proteins’ (RBPs) transcriptional and translational regulation roles in human heart. The authors re-analyzed previously published paired Ribo-seq and RNA-seq data from 80 human hearts and interrogated the correlation between expression levels of the 143 cardiac-expressed RBPs and the expression level and translational efficiency (TE) of their corresponding RNA-binding targets (obtained from ENCODE database, CLIP-seq approaches). In total, 58 RBPs were found to have significant associations with target mRNA abundance (denoted as “mRNA-RBPs”) and 37 RBPs with significant associations with target translational efficiencies (denoted as “TE-RBPs”), among which 21 RBPs were found in both categories and denoted as multifunctional RBPs. These includes 27 novel TE-RBPs, previously unknown for translational control process, including a muscle-specific and dilated cardiomyopathy-associated splicing regulator RBM20. The authors showed the multifunctional RBPs are context-specific where their functional outcome depends on the set of mRNAs they target, and their target genes also appear to be regulated independently, facilitated by a differential affinity for coding sequence length or 5’ UTR structure. In summary, this study gives a comprehensive analysis on the RBP regulatory roles in human heart based on current available transcriptome, translational efficiency and RNA-binding data with rigorous experiment design and statistical tests. It provides a list of potential TE-RBPs, multifunctional RBPs and many interesting insights of how these RBPs regulating the expression and translation, which could be further validated and studied in this field.

Major Comments:

1. This study comes up with a very interesting sets of RBPs that potentially participate in the expression and translation control of the mRNAs and provides the directionality information as well. I was wondering how this list overlaps with previously published screen studies that focused on the RBP functions and that also showed directionality, such as the study Luo et al. in Nature Structural & Molecular Biology (2020) and other similar studies.

Minor comments:

1. In the “Multifunctional RBPs including DDX3X and G3BP1 regulate mRNA abundance and translational efficiency of independent sets of target genes” of Results section, I have some problem in understanding the expression “which decreased further for the most strongly correlating targets”( line 11 of the paragraph) and Figure S3 A. I was wondering if the author could explain a bit more, highlight the “most strongly correlating targets” in the corresponding figure or something that can help readers better understand the results.

2. In the discussion, the authors mentioned all discovered multifunctional RBPs are ubiquitously expressed across human tissues, but I did not find the data or plots presenting this result in the manuscript. I suggest it would be great if the authors could show the results in the main figures or supplementary figures (e.g. using heatmap and GTEx data).

**Have the authors made all data and (if applicable) computational code underlying the findings in their manuscript fully available?**

Reviewer #1: Yes

Reviewer #2: Yes

Reviewer #3: Yes

PLOS authors have the option to publish the peer review history of their article (what does this mean?). If published, this will include your full peer review and any attached files.

Reviewer #1: No

Reviewer #2: No

Reviewer #3: No

Figure Files:

Data Requirements:

Reproducibility:

References:

---

## [Editor Report · Decision Letter 1]

18 Nov 2021

Dear van Heesch,

We are pleased to inform you that your manuscript 'Multifunctional RNA-binding proteins influence mRNA abundance and translational efficiency of distinct sets of target genes' has been provisionally accepted for publication in PLOS Computational Biology.

Best regards,

Greg Tucker-Kellogg, PhD

Associate Editor

PLOS Computational Biology

Daniel Beard

Deputy Editor

PLOS Computational Biology

---

## [Editor Report · Acceptance letter]

29 Nov 2021

PCOMPBIOL-D-21-01425R1 

Multifunctional RNA-binding proteins influence mRNA abundance and translational efficiency of distinct sets of target genes

Dear Dr van Heesch,

I am pleased to inform you that your manuscript has been formally accepted for publication in PLOS Computational Biology. Your manuscript is now with our production department and you will be notified of the publication date in due course.

With kind regards,

Agnes Pap
